# Oligomerization Affects the Ability of Human Cyclase-Associated Proteins 1 and 2 to Promote Actin Severing by Cofilins

**DOI:** 10.3390/ijms20225647

**Published:** 2019-11-12

**Authors:** Vedud Purde, Florian Busch, Elena Kudryashova, Vicki H. Wysocki, Dmitri S. Kudryashov

**Affiliations:** 1Department of Chemistry and Biochemistry, The Ohio State University, Columbus, OH 43210, USA; purde.1@osu.edu (V.P.); busch.151@osu.edu (F.B.); kudryashova.1@osu.edu (E.K.); wysocki.11@osu.edu (V.H.W.); 2The Ohio State Biochemistry Program, The Ohio State University, Columbus, OH 43210, USA; 3Resource for Native MS-Guided Structural Biology, The Ohio State University, Columbus, OH 43210, USA; 4Campus Chemical Instrument Center, Mass Spectrometry and Proteomics, The Ohio State University, Columbus, OH 43210, USA

**Keywords:** cyclase-associated proteins, oligomerization, coiled coils, actin severing, actin depolymerization, α-barrels

## Abstract

Actin-depolymerizing factor (ADF)/cofilins accelerate actin turnover by severing aged actin filaments and promoting the dissociation of actin subunits. In the cell, ADF/cofilins are assisted by other proteins, among which cyclase-associated proteins 1 and 2 (CAP1,2) are particularly important. The N-terminal half of CAP has been shown to promote actin filament dynamics by enhancing ADF-/cofilin-mediated actin severing, while the central and C-terminal domains are involved in recharging the depolymerized ADP–G-actin/cofilin complexes with ATP and profilin. We analyzed the ability of the N-terminal fragments of human CAP1 and CAP2 to assist human isoforms of “muscle” (CFL2) and “non-muscle” (CFL1) cofilins in accelerating actin dynamics. By conducting bulk actin depolymerization assays and monitoring single-filament severing by total internal reflection fluorescence (TIRF) microscopy, we found that the N-terminal domains of both isoforms enhanced cofilin-mediated severing and depolymerization at similar rates. According to our analytical sedimentation and native mass spectrometry data, the N-terminal recombinant fragments of both human CAP isoforms form tetramers. Replacement of the original oligomerization domain of CAPs with artificial coiled-coil sequences of known oligomerization patterns showed that the activity of the proteins is directly proportional to the stoichiometry of their oligomerization; i.e., tetramers and trimers are more potent than dimers, which are more effective than monomers. Along with higher binding affinities of the higher-order oligomers to actin, this observation suggests that the mechanism of actin severing and depolymerization involves simultaneous or consequent and coordinated binding of more than one N-CAP domain to F-actin/cofilin complexes.

## 1. Introduction

A dynamic balance between the polymerization of actin filaments at the barbed ends and depolymerization at the pointed ends is crucial for numerous cellular processes including cell adhesion, cell motility, cytokinesis, and morphogenesis [1,2,3,4]. To ensure a prompt response to cell needs and environmental challenges, fast rates of actin polymerization are underpinned by several mechanisms conferring high local concentrations of monomeric actin at polymerization sites [5,6]. To support these fast polymerization rates, prompt replenishing of the G-actin pool should be balanced by equally fast depolymerization of old filaments [2,3]. Fast depolymerization is achieved (1) via a severing-mediated increase in the number of the depolymerizing filament ends and (2) by promoting dissociation of actin subunits from the ends. Actin-depolymerizing factor (ADF)/cofilin family proteins play a key role in both mechanisms [3,7,8].

There are three isoforms of ADF/cofilins in mammals. Actin depolymerizing factor (ADF) is highly expressed in epithelial and endothelial cells; non-muscle cofilin 1 (CFL1) is expressed ubiquitously; and cofilin 2 (CFL2) is found predominantly, but not exclusively, in muscle tissues [8]. By selectively recognizing conformational changes in actin following ATP hydrolysis and P_i_ release [9,10], ADF/cofilins favorably decorate aged regions of F-actin, ensuring their prevalent disassembly. Binding of ADF/cofilins changes the filament twist [11] and reshapes the contacts between actin subunits [12,13,14,15,16,17], which results in severance of the borders between cofilin-decorated and non-decorated filament regions [18]. Since every severing event produces not only a new pointed end prone to depolymerization, but also a barbed end capable of elongation, the severing ability of cofilin was questioned initially as non-physiological [19,20,21]. This issue was resolved, however, by the discovery of proteins that assist ADF/cofilins in the controlled acceleration of actin depolymerization: actin-interacting protein 1 (Aip1), coronin, twinfilin, and cyclase-associated protein (Srv2/CAP) [22,23,24,25,26]. Working together, these proteins enable the mechanisms of selective nucleotide-dependent depolymerization of aged filaments [25,27], prevent polymerization at the barbed ends formed as a result of severing [26], accelerate dissociation of actin subunits from severed filaments [24], and stimulate dissociation of ADP from the depolymerized G-actin and its recharging by ATP [28]. Of these proteins, cyclase-associated proteins Srv2/CAPs [22,29,30] are of particular interest due to their multi-faceted contribution to the depolymerization process. The binding of Srv2/CAPs to cofilin-decorated actin via its N-terminal helical folded domain (HFD) potentiates cofilin-mediated severing [31] and, with the assistance of twinfilin, promotes disassembly of F-actin 3- and 17-fold at the barbed and pointed ends, respectively [32]. The central region of the protein contains a WH2 domain sandwiched between two proline-rich motifs (P_1_ and P_2_), which is followed by a C-terminal β-strand folded domain (Figure 1a) [33,34]. Binding of WH2 and β-strand domains to the ADP–G-actin/cofilin complex produced as a result of depolymerization promotes dissociation of both cofilin and ADP, and facilitates the recharging of the monomers with ATP [28]. The newly formed ATP–G-actin monomer is either released or actively transferred to profilin, thus completing the goal of regenerating the pool of polymerization-competent actin [28,35,36,37]. 

The full-length Srv2/CAP protein (FL CAP) has been found by gel filtration chromatography and analytical ultracentrifugation to form hexamers, while the N-terminal half (N-CAP) has presented a six-bladed shuriken shape in negative staining electron microscopy reconstructions [29,31,38]. The oligomerization is functionally important, and the N-terminal α-helix is thought to be critical for the formation of the hexamers, as its removal reduced the ability of the HFD domain to potentiate severing and depolymerization [31,38]. However, the role of Srv2/CAP oligomerization in this potentiation is not understood. Furthermore, the structure of the Srv2/CAP oligomerization is unclear. Approximately 98% of known coiled coils form low-order oligomers from dimers to tetramers, while the occurrence of natural coiled elements of a higher order is very rare [39] for reasons that are not well understood.

The two human isoforms of cyclase associated protein (CAP1 and CAP2) share 64%/75% sequence identity/similarity with each other (Figure 1b) and ~35%/50% identity/similarity with yeast Srv2 [40]. CAP1 is expressed ubiquitously, whereas CAP2 is produced predominantly in striated muscles, but is also found at lower levels in brain, testes, lung, liver, and skin tissues [40,41]. Accordingly, the most notable outcome of *CAP2* gene ablation in mouse is cardiomyopathy [42]. At the cellular level, CAP1 is found exclusively in the cytoplasm, while CAP2 shows dual localization in the cytoplasm and the nucleus [41]. 

Both human CAP isoforms have recently emerged as markers of invasive tumors. Thus, CAP1 has been implicated in breast, lung, pancreatic, and ovarian cancers along with glioma, hepatocellular, and head and neck squamous cell carcinomas [43,44,45,46]. CAP2 is overproduced in hepatocellular carcinoma, malignant melanoma, breast, and gastric cancers [47,48], and its expression is associated with poor clinical outcomes [44,49]. CAP2 is also overproduced by bladder, colon, kidney, and thyroid tumors [41], making it a common marker for various tumors. However, the exact role of CAPs in tumorigenesis is ambiguous, and anti-oncogenic action of the CAP1 and 2 isoforms for other tumors has also been proposed [45]. Interestingly, cancer-linked mutations in both CAP1 and CAP2 are enriched in the N-terminal domain, with the hot spot mutations in CAP1 are located at the Arg-29 residue [44], which is not conserved between the two isoforms. While mouse CAP1 (95% identical to its human counterpart) has been partially characterized [38], the mammalian CAP2 isoform or its fragments have never been biochemically explored. Therefore, the goals of the current study were (1) to evaluate and compare the abilities of the N-terminal fragments of the two human CAP isoforms to potentiate actin severing and depolymerization by human cofilins and (2) to explore how the oligomerization stoichiometry affects these abilities.

By conducting bulk actin depolymerization assays and monitoring single-filament severing using total internal reflection fluorescence (TIRF) microscopy, we found that the N-terminal domain of both isoforms (N-CAP) enhanced cofilin-mediated severing and depolymerization to a similar degree. Surprisingly, our analytical sedimentation and native mass spectrometry data showed that the N-terminal fragments of both human CAP isoforms formed tetramers rather than the hexamers reported for full-length Srv2 and mouse CAP1 [31,38]. By replacing the original oligomerization domains of CAPs with artificial coiled-coil sequences of known oligomerization patterns [50], we found that the activity of the proteins correlates with the stoichiometry of their oligomerization; i.e., tetramers and trimers more potently promoted severing and depolymerization of actin by both cofilin isoforms than dimers and monomers. 

## 2. Results

### 2.1. Analytical Ultracentrifugation and Native Mass Spectrometry Reveaedl that the N-terminal Domains of Recombinant Human CAPs Form Tetramers

The N-terminal domain of N-Srv2 and mammalian CAPs contains a coiled-coil region followed by the helical folded domain (HFD) (Figure 1a). HFD interacts with F-actin and enhances cofilin-mediated F-actin disassembly [31,38], while the coiled coil contributes to oligomerization [38]. Specifically, the HFD of Srv2 has been proposed to form hexamers with radial symmetry and a shuriken-(star)-like appearance [31]. The extreme N-terminus of Srv2/CAP has been predicted to form a coiled-coil helix, but whether it can dictate the quaternary structure of CAPs is unclear, as coiled coils with a number of chains higher than four are rare in nature. We employed sedimentation velocity analytical ultracentrifugation (SV-AUC) to determine the oligomerization states of the recombinant constructs of human CAP1 and CAP2 HFDs with the original N-terminal coiled coils (N-CAP1 and N-CAP2) and 6xHis-tags placed at the C-termini to avoid interference with oligomerization. Raw AUC data (Appendix A) were analyzed using the SEDFIT analysis tool [51], and the molecular weights (MW) of the analyzed constructs were calculated based on their sedimentation coefficient values (Figure 2a and Table 1). 

Surprisingly, the AUC-determined MWs of both N-CAP1 and N-CAP2 constructs with the original N-terminal coiled coils matched those of tetramers, and not hexamers as has been suggested for yeast Srv2 and mouse CAP1 counterparts [31,38]. To validate the unexpected results, we analyzed the oligomerization state of each construct using native mass spectrometry (native MS; Figure 2b and Appendix A) and confirmed that the main oligomeric state of both N-CAP1 and N-CAP2 constructs was tetrameric, with a minor presence of the lower oligomerization state species (Figure 2 and Table 1). Varying the buffer (Tris(hydroxymethyl)aminomethane (Tris)–HCl vs. phosphate-buffered saline (PBS)) and salt (30 mM NaCl vs. 135 mM KHPO_4_) composition did not affect the oligomerization state of the proteins. 

In an attempt to resolve the difference between the reported data and our results, we computationally analyzed the ability of the CAP N-terminal helix to form coiled coils using CCBUILDER 2.0 software [52], which was created for prediction and de novo design of high-order α-helical oligomers. CCBUILDER predicted that the N-terminal helices of CAP1 and CAP2 (Figure 3) were substantially more likely to form hexamers than tetramers, as reflected in over two-fold higher negative energy amplitudes for interface packing, expressed in the BUDE (Bristol University Docking Engine) force field format [52] (Table 2). It is worth noticing, however, that heptamers of both CAP1 and CAP2 showed even lower BUDE energies, pointing to potential limitations of the approach. 

Analysis of the best-scored model structures showed that the tetramers followed a classical Type I “hxxhhxx” pattern (where “h” is a hydrophobic residue), with two “h” residues stabilizing neighboring partners and one shared between the hydrophobic coil-stabilizing seams (Figure 3b,c,g,h,l). Oligomers formed by five or more α-helices are called α-barrels as they have an internal cavity or channel. With the increased diameter, hexameric coiled-coil α-barrels require more hydrophobic residues to stabilize the core, and their residues more commonly follow Type II “hhxxhhx” heptad patterns [39]. The CCBUILDER model predicted that the role of the fourth hydrophobic residue in the heptad, in this case, would be played by the Cβ-Cγ atoms of the Arg residues, the charged side chains of which would contribute to stabilization of the helical barrels via salt bridge interactions (yellow dashed lines in Figure 3f) with Asp and Glu residues of the neighboring helices.

### 2.2. Construction of Monomers, Dimers, and Trimers of Human CAP1 and CAP2 HFD Domains

To address the role of the stoichiometry of N-CAP oligomers in their ability to enhance cofilin-mediated actin depolymerization, we generated recombinant constructs of the HFD domains with the native coiled-coil helices either deleted (∆CC-N-CAPs) or replaced by helical elements with known dimer (DD-N-CAPs) and trimer (TD-N-CAPs) oligomerization stoichiometry [50] (Figure 1a). 

The MW of the ΔCC-N-CAP1 and ΔCC-N-CAP2 constructs, experimentally determined by AUC, matched the theoretical MW of the respective constructs with 93 and 99% accuracy (Figure 2a and Table 1). The constructs designed as dimers and trimers matched the theoretical expectations with 95% and 94% accuracy, respectively (Figure 2a and Table 1). Furthermore, native MS confirmed the desired oligomerization states of the constructs with high accuracy, while also showing the presence of a minor population of dimers of both ∆CC-N-CAPs (Figure 2b and Table 1). The latter observation tentatively suggests that HFD may contribute to the interactions between the subunits and stabilization of the native quaternary state of the CAP oligomers. 

### 2.3. Oligomerization through the N-terminal Coiled-Coil Region Affects the Binding Affinity of N-CAPs to F-actin

Assuming that the ability of Srv2 to enhance cofilin-mediated actin severing requires binding to F-actin via the N-terminal HFD domain [31], we tested the effects of oligomerization on the ability of human CAP1 and CAP2 HFD constructs to bind F-actin. By titrating a fixed concentration of N-CAP1 construct with various concentrations of F-actin, we found that the affinity of N-CAP1 constructs steadily increased from monomers to tetramers (Figure 4b–e,g). This higher affinity suggested that more than one subunit of the oligomers can contribute to the interaction with F-actin in natural and artificially designed N-CAP oligomers. Although N-CAP2 constructs showed a similar trend, the affinity for the monomeric form was lower, while the affinity of the tetrameric constructs was higher than that of N-CAP1 constructs (Figure 4f,g).

### 2.4. Higher Affinities of the N-CAP Constructs to F-actin Correlated with Higher Potentiation of F-actin Depolymerization by Cofilins

To assess whether the increased affinity correlated with an improved function, we tested the ability of the N-CAP constructs to enhance cofilin-mediated F-actin disassembly in bulk pyrene–actin depolymerization assays. Given that the isoforms of human cofilins differ in their properties [8,53] and both CFL2 and CAP2 are predominantly found in muscle tissues, while both CFL1 and CAP1 are expressed ubiquitously, our goal was to establish whether the pairs of cofilin and CAP isoforms showed functional selectivity. We observed that in both the absence and presence of N-CAP constructs, depolymerizetion by CFL1 was overall more effective than that by CFL2 (Figure 5, Table 3, and Appendix A). The concentrations of all the oligomers in these experiments were normalized to their monomeric forms, so the actual concentrations of the protein complexes were inversely proportional to their oligomerization state. Nonetheless, both N-CAP1 and N-CAP2 constructs potentiated actin depolymerization progressively better with increasing level of oligomerization, suggesting that proximity of the subunits in the oligomers positively contributed to the mechanism of severing and depolymerization. However, while dimers were notably more active than monomers, trimers were nearly as effective as tetramers, with the only exception being the N-CAP2 dimer, which was as effective as the CAP2 trimers and tetramers while cooperating with CFL2 (Appendix A and Table 3). There was no difference between the functional efficiencies of the CAP isoforms when assisting CFL1, and CAP2 was only marginally more effective than CAP1 when potentiating depolymerization by CFL2 (Figure 5, Appendix A, and Table 3). 

To separate the effects due to the potentiation of severing from those caused by the acceleration of depolymerization from the ends, we analyzed severing effects at the single-filament level using TIRF microscopy. To this end, fluorescently labeled G-actin (33% Alexa 488-labeled actin and 1% biotinylated actin) was polymerized in a chamber and tethered to coverslips through biotin–streptavidin interactions. Filaments were grown up to 15 µm average size before either cofilin alone (50 nM) or with N-CAP constructs (750 nM) was flowed into the chamber to induce severing. The applied proteins did not contain G-actin to preclude polymerization from the severed ends of the filaments. As compared to CFL1 alone, we observed over 2.5-fold potentiation of severing in the presence of either N-CAP1 or N-CAP2 (Figure 6a–c) with no substantial difference between the two isoforms. Again, the severing events directly correlated with the N-CAP oligomeric stoichiometry, with the tetrameric constructs being notably more effective than their lower oligomerization state counterparts (Figure 6b,c). Interestingly, despite slower depolymerization in bulk assays, CFL2 severed filaments more effectively than CFL1 (Figure 6b–d). The severing was so fast and effective that both N-CAP constructs contributed only marginally to the severing activity of CFL2 (Figure 6d). These observations are in line with a higher severing efficiency of CFL2 towards skeletal actin [53,54] and its lower depolymerization capacity [8,54] reflected in its inability to increase the critical concentration for polymerization [17].

### 2.5. Interaction of CAP1 with CFL1 and CFL2 Isoforms in Cells

To check whether the isoforms of cofilin and CAP made preferential interactions in cells, we employed the rolling cycle amplification-based proximity ligation assay (Duolink In Situ PLA), allowing detection of interacting partners when they are located within ~40 nm from each other (see Section 4.8). By testing various commercial antibodies, we were able to identify specific antibodies that recognized both recombinant proteins and a single band of the correct size for CFL1, CFL2, and CAP1 (Figure 7a and Appendix A). Unfortunately, CAP2 antibodies either detected a single band but of the wrong size in cell extracts and failed to detect recombinant CAP2 (Santa Cruz Biotechnology #sc-377471; Appendix A), or detected the recombinant protein and the right size protein in cell lysates, but only as a weak-intensity band among about a dozen higher-intensity bands (Abnova #H00010486-M01; Appendix A), which made the antibody inappropriate for immunocytochemistry (and PLA, in particular). To explore the localization of CAP1 with both isoforms of cofilin, we screened over two dozen cell lines (Appendix A) and selected Hs 578T breast epithelial carcinoma cells, which express high levels of both CFL1 and CFL2 (Figure 7a). Despite CFL1 being the more prevalent of the two isoforms in these cells, the CFL2/CAP1 pair generated notably more proximity ligation events (Figure 7b), suggesting their more prominent co-localization and interaction. Notice, however, that more prominent localization may result from the low depolymerization ability of CFL2 (Table 3), leading to more prolonged, but functionally less effective, cooperation.

## 3. Discussion

Elevated expression of both human CAP isoforms (CAP1 and CAP2) is linked to poor prognosis in multiple different metastatic cancers [45,55]. High invasiveness of cancer cells overexpressing CAP1 and CAP2 correlates with their roles in stimulating the turnover of actin filaments, which is essential for cell motility via promoted formation and turnover of pro-migratory structures such as filopodia and lamellipodia, leading to metastasis [45]. Since cancer-related mutations in both CAP1 and CAP2 are more prevalent in the N-terminal part of the CAP isoforms, we compared the biochemical properties of the N-terminal region of both isoforms, which are recognized to bind F-actin and enhance cofilin-mediated filament disassembly.

Although native Srv2 and CAP1 are reported to form hexamers, as originally revealed by gel filtration and reconstruction of negatively stained electron microscopy images [29,38] and recently confirmed by AUC [56], we found that the recombinant N-terminal regions of human CAP1 and CAP2, that contained the coiled-coil helix and HFD domains both formed tetramers. A computational approach also suggested that for the N-terminal coiled coils of both CAPs, hexamers are more enthalpically favorable than tetramers (Figure 3 and Table 2). Several factors could contribute to this discrepancy. First, in contrast to tetramers, hexamers were predicted to have an internal channel inlaid by Leu side chains and large enough to incorporate water (Figure 3g,j). Such channels would be entropically disfavored unless occupied by nonpolar molecules, e.g., fatty acids. Such complexes with fatty acids, trans-retinol, and Vitamin D have been demonstrated for the pentameric α-barrel of cartilage oligomeric matrix protein (COMP) [57,58]. We noticed that saturated fatty acids in their extended conformation fit well in the central cavity of the CAP1/2 models (Figure 3k). Therefore, it is appealing to consider that, under physiological conditions of the cell, CAP hexamers are stabilized by such nonpolar molecules. Next, the hexamers of N-CAPs can be disfavored by the lower configurational entropy of HFD domains, which were expected to be constrained in hexamers but loosely placed in lower-level oligomers. In the full-length proteins, the equilibrium can be shifted towards the hexameric state by the enthalpic forces of the pairwise association between the CAP C-terminal domains [28]. One could also speculate that the formation of hexamers or other high-order constitutive oligomers in the cell occurs co-translationally on polyribosomes, as has been shown for the assembly of vault particles [59]. This mechanism could also reduce non-productive or mistargeted interactions with other proteins and disfavor the formation of hetero-hexamers of CAP1 and CAP2. In the absence of such mechanisms, the proposed hetero-hexamers are likely upon simultaneous expression of the CAP1 and CAP2 isoforms, due to the high conservation of the coiled-coil domains varying by only four conserved amino acids (Figure 1b).

Although the stoichiometry of the recombinant CAP constructs did not match the reported stoichiometry of the native complexes, the recombinant tetramers were nearly 2-fold more effective in assisting cofilin in actin severing and depolymerization as compared to the HFD monomers (Figure 5 and Figure 6). Furthermore, oligomers of a reduced stoichiometry (i.e., trimers and dimers) also assisted cofilin better than monomers (Appendix A). Along with the higher binding affinities of the oligomers to actin (Figure 4g), this observation suggested that the mechanism of actin severing and depolymerization involves simultaneous or consequent and coordinated binding of more than one HFD domain to F-actin–cofilin complexes. It is tempting to speculate that the CAP oligomers may roll at the depolymerizing end of F-actin as a wheel, with “spokes” of HFD binding to both terminal and adjacent actin subunits, separating the former from the latter and from the filament body. Since severing occurs at the interface between the cofilin-decorated and cofilin-free filament regions characterized by different twists, the HFD domains of CAPs may serve to stabilize the original actin twist and provide a distinct border between the two states.

We did not observe a measurable difference between the CAP isoforms in their ability to assist in actin depolymerization (Figure 5). This observation correlates with the fact that both CAP1 and CAP2 are recognized as makers of tumorigenesis, promoting invasiveness and cytokinesis of tumor cells. While more effective in severing (Figure 6b–d), CFL2 was notably less potent than CFL1 in accelerating filament depolymerization in bulk assays (Table 3). This difference between CFL1 and CFL2 persisted in the presence of N-CAP1/2 constructs, and may account for the more abundant association of CFL2 with CAP1 observed in cells (Figure 7b). Indeed, slower dissociation of actin subunits from the filament ends might imply cycling via less productive but more extended in time association of CFL2 with CAPs. Since our experiments did not reveal a substantial functional difference between N-CAP1 and N-CAP2, it is conceivable that the difference between the isoforms is limited to their differential regulation and/or their C-terminal regions involved in recharging the depolymerized ADP–G-actin/cofilin complexes with ATP and profilin [28].

In addition to promoting actin turnover, ADF/cofilins mediate active, importin-mediated transport of actin to the nucleus [60,61,62], stabilize thick parallel F-actin bundles (rods) both in the cytoplasm and the nucleus under stress conditions [63,64,65], regulate mitochondrial dynamics [66,67,68], and, controversially, contribute to apoptosis upon translocation to mitochondria [69,70,71,72,73]. These activities appear to correlate with a low energy state or a decreased cell’s reducing power, and are designed to either compensate for these deficiencies by reducing actin treadmilling and the resulted energy consumption (e.g., by sequestering ADP–actin in actin rods) or exacerbating the condition by promoting cell death (e.g., apoptosis). Nuclear transport of actin may serve similar compensatory purposes, as high levels of nuclear actin inhibit transcription by RNA polymerase II [74]. Whether or not CAP1/CAP2 assist cofilin in these processes is unknown and remains to be established in future works. 

## 4. Materials and Methods

### 4.1. Molecular Cloning

N-CAP constructs with the original coiled-coil N-terminal helices were cloned with C-terminal 6xHis-tags into pET21b plasmid (Novagen, Madison, WI, USA.). Dimerization 2L4HC2_23 (5J0K), and trimerization 2L6HC3_6 (5J0J) domain sequences [50] were amplified from plasmids generously provided by Dr David Baker (University of Washington, Seattle, WA, USA). Coiled-coil regions of N-CAPs (a.a. 1–29) were deleted (to generate ΔCC-N-CAP constructs) or replaced with either dimerization or trimerization domains (to generate DD-N-CAPs and TD-N-CAPs, respectively) and cloned with N-terminal 6×His-tags into pColdI vector (Clontech, San Francisco, CA, USA) using NEBuilder HiFi DNA Assembly Master Mix (New England BioLabs, Ipswich, MA, USA). Sequences of all constructs were verified by Sanger DNA sequencing (Genomics Shared Resource, OSU Comprehensive Cancer Center, Columbus, OH, USA).

### 4.2. Protein Purification

Actin was purified from skeletal muscle acetone powder: either of rabbit origin purchased from Pel-Freez Biologicals (Rogers, AR, USA) or of chicken origin prepared in-house from flash-frozen chicken breast (Trader Joe’s, Columbus, OH, USA), as previously described [75,76,77].

CAP1 and CAP2 constructs (N-CAP, Δ-CC-CAP, DD-N-CAP, TD-N-CAP) were expressed in BL21-CodonPlus(DE3)pLysS *Escherichia coli* (Agilent Technologies, Santa Clara, CA, USA) grown in nutrient-rich bacterial growth medium (1.25% tryptone, 2.5% yeast extract, 125 mM NaCl, 0.4% glycerol, 50 mM tris(hydroxymethyl)aminomethane hydrochloride (Tris-HCl), pH 8.2). After reaching an OD_600_ of 1–1.2, the cells were incubated for 30 minutes on ice, and expression was induced by the addition of 1 mM isopropyl-β-d-thiogalactoside (IPTG), after which the cells were grown at 15 °C for 15–20 hours. CAP constructs were purified by immobilized metal affinity chromatography on TALON metal affinity resin (Clontech, San Francisco, CA, USA) and eluted in buffer containing 50 mM sodium phosphate, pH 7.4, 300 mM NaCl, 0.1 mM phenylmethylsulfonyl fluoride (PMSF), 2 mM benzamidine–HCl, and 250 mM imidazole. Purified constructs were dialyzed against buffer containing 20 mM Tris-HCl, pH 8.0, 30 mM NaCl, 2 mM dithiothreitol (DTT), and 0.1 mM PMSF. CAP constructs were aliquoted, flash-frozen in liquid nitrogen, and stored at −80 °C.

Tagless full-length human cofilins were expressed in *E. coli* BL21-CodonPlus(DE3) cells. Transformed bacterial cells were grown at 37 °C in 4 L of a rich medium as described above. Cultures were induced with 1 mM IPTG, and incubated at 37 °C in a shaking incubator for 4 h. Cells were pelleted at 4 °C and resuspended in ice-cold buffer containing 10 mM piperazine-N,N′-bis(2-ethanesulfonic acid) (PIPES), pH 6.8, 0.5 mM ethylenediaminetetraacetic acid (EDTA), 10 mM 2-mercaptoethanol, 1 mM PMSF, 5 mM benzamidine, SIGMAFAST protease inhibitor cocktail (EDTA-free), and lysed using a French press. Cofilins were purified by sequential anion and cation exchange chromatography as described previously [78]. Briefly, cell lysates were passed through DE52 (DEAE cellulose) and suplphopropyl (SP)-sepharose (Sigma-Aldrich, St. Louis, MO, USA) columns connected in tandem (in this order), followed by elution from the SP-sepharose column with a gradient of 50 to 500 mM NaCl in buffer containing 10 mM PIPES, pH 6.8, 0.5 mM EDTA, 10 mM 2-mercaptoethanol, and 0.5 mM PMSF. The recombinant cofilins were purified to >95% homogeneity by size-exclusion fast protein liquid chromatography (FPLC) in 10 mM 3-(N-morpholino)propanesulfonic acid (MOPS), pH 7.0, 25 mM NaCl, 1 mM DTT, and 0.1 mM PMSF.

Concentrations of all proteins were determined based on their absorption: for actin in the presence of 0.5 M NaOH, an *A* (1%) at 290 nm of 11.5 cm^−1^ was assumed; extinction coefficients at 280 nm for CFL1 and CFL2 as well as for all N-CAP1 and N-CAP2 constructs were predicted based on their sequences using the Expasy ProtParam online resource, Switzerland [79].

### 4.3. Sedimentation Velocity Analytical Ultracentrifugation (SV-AUC)

Sedimentation velocity experiments were performed in a ProteomeLab XL-I analytical ultracentrifugation system (Beckman Coulter, Chaska, MN, USA). Briefly, 50 µM protein samples were loaded into AUC cell assemblies at a 12 mm path length. To achieve chemical and thermal equilibrium, the An-50 TI rotor with loaded samples was allowed to equilibrate in the centrifuge at 20 °C for ~4 h. The rotor was spun at 50,000 rpm and absorption at 280 nm data were collected for up to 6 hours and a total of 42 scans. SEDFIT software (http://sedfitsedphat.nibib.nih.gov, version 16-1c, USA) was used to perform the data analysis with a continuous sedimentation coefficient distribution model *c*(*S*). Values for 20 mM Tris-HCl, pH 8.0 buffer viscosity (0.010102 poise), density (1.02 g/mL), and partial specific volume (0.73 mL/g) were used, and confidence level was set to 0.68.

### 4.4. F-Actin Binding Cosedimentation Assays

Ca^2+^ in the nucleotide cleft of G-actin was switched to Mg^2+^ by adding MgCl_2_ and ethylene glycol-bis(β-aminoethyl ether)-N,N,N′,N′-tetraacetic acid (EGTA) to 0.1 and 0.5 mM, respectively, and incubating on ice for 10 min. G-actin was then polymerized by supplementing MgCl_2_ and KCl to 2 and 50 mM, respectively, in 20 mM Tris-HCl, pH 7.5, and incubating at room temperature for 1 h. CAPs were used at a final concentration of 5 μM, while actin concentration varied from 2.5 to 50 μM. Reaction mixtures were incubated either 1 h at room temperature or overnight at 4 °C. Reactions were centrifuged at 300,000× *g* at 25 °C for 30 min using a TLA-100 rotor in an Optima TLX ultracentrifuge (Beckman Coulter, Chaska, MN, USA). Supernatants and pellets were carefully separated, balanced by volume, and analyzed by SDS-PAGE. Gels were stained with Coomassie Brilliant Blue and quantified using ImageJ software version 2.0.0.-rc-69/1.52p, USA [80]. Binding efficiency expressed as an equilibrium dissociation constant (K_d_) was quantified by fitting the data to the binding isotherm equation
$$\Delta F/\Delta F_{max}=(P+A+K_{d}-\sqrt{{\left((P+A+K_{d}\right)}^{2}-4PA)}/2P$$where *A* is the concentration of F-actin and *P* is the concentration of N-CAP constructs.

### 4.5. TIRF Microscopy

TIRF microscopy was conducted as described previously [76,77]. Briefly, immediately before the experiment, flow chambers were functionalized by incubation with 0.1 mg/ml streptavidin in phosphate-buffered saline (PBS) and blocked for 3 min in blocking buffer (1% (*w*/*v*) bovine serum albumin (BSA) in 50 mM Tris-HCl, pH 7.5, 150 mM NaCl), followed by successive washes with the blocking buffer and 1× TIRF buffer (10 mM imidazole, pH 7.0, 50 mM KCl, 50 mM DTT, 1 mM MgCl_2_, 1 mM EGTA, 0.2 mM ATP, 50 µM CaCl_2_, 15 mM glucose, 20 µg/mL catalase, 100 µg/mL glucose oxidase, 15% glycerol, and 0.5% methylcellulose-400cP (Sigma Aldrich, St. Louis, MO, USA)). Skeletal actin (33% Alexa 488-labeled, 1% biotinylated; 1.5 µM final concentration) was incubated with an exchange buffer (50 µM MgCl_2_, 0.1 mM EGTA) for 2 min in order to switch from Ca^2+^– to Mg^2+^–ATP actin. Polymerization of actin was initiated by the addition of an equal volume of 2× TIRF buffer and the mixture was transferred to the flow chamber within 15 seconds. Filaments were grown to ~15 µM average length. Free actin monomers were then removed by flushing the desired concentrations of proteins in 1× TIRF buffer. Images were collected every 5 s with a Nikon Eclipse Ti-E microscope, through-the-objective TIRF illumination system, 100× oil objective, and a DS-QiMc camera (Nikon Instruments Inc., Melville, NY, USA). Data were analyzed using ImageJ software: severing events per µM of filament length were calculated by measuring the filament length in the frame prior to the flow of proteins and then manually counting the number of severing events.

### 4.6. Bulk F-actin Disassembly Assays

The final concentration of 2 µM, 10% pyrene-labeled F-actin was mixed with 4 µM latrunculin B and 100 nM CapZ in F-buffer (20 nM Tris-HCl, pH 7.5, 50 mM KCl, 0.2 mM ATP, 1 mM MgCl_2_, 0.5 mM EGTA, and 1 mM DTT). At time zero, disassembly was induced by addition of either F-buffer alone, F-buffer containing cofilin, or F-buffer containing cofilin and CAPs. A decrease in fluorescence was monitored for 3000 s at 25 °C at excitation wavelength (λex = 365 nm) and emission wavelength (λem = 407 nm) using an Infinite M1000 Pro plate reader (Tecan, Baldwin Park, CA, USA).

### 4.7. Native Mass Spectrometry (Native MS)

Sample purity and integrity were analyzed by online buffer exchange MS using an UltiMate™ 3000 RSLC coupled to an Exactive Plus EMR Orbitrap instrument (Thermo Fisher Scientific, Grand Island, NY, USA) modified to incorporate a quadrupole mass filter and allow for surface-induced dissociation [81]. Between 100 and 300 pmole protein (referring to monomer) were injected and online buffer exchanged to 200 mM ammonium acetate, pH 6.8 by a self-packed buffer exchange column [81,82] (P6 polyacrylamide gel; Bio-Rad Laboratories, Hercules, CA, USA) at a flow rate of 100 µL per min. Mass spectra were recorded for 1000–14000 *m*/*z* at 8750 resolution, as defined at 200 *m*/*z*. The injection time was set to 200 ms. Voltages applied to the transfer optics were optimized to allow ion transmission while minimizing unintentional ion activation. Mass spectra were deconvoluted with UniDec version 4.0.0 beta, England [83].

### 4.8. Cell Culture, Western Blotting, and In Situ Proximity Ligation Assay (PLA)

To reveal the native interactions of CAP proteins with cofilin isoforms in cells, we utilized a Duolink in situ proximity ligation immunofluorescence assay (Sigma-Aldrich, St. Louis, MO, USA). In this assay, two primary antibodies raised in different species are used to detect two unique protein targets (e.g., CFL and CAP) followed by binding of the corresponding species-specific secondary antibodies conjugated with oligonucleotides (PLA probes). If the target proteins interact with each other, the corresponding PLA probes will be close to each other (<40 nm) and hybridizing connector oligos will join the PLA probes, consequently amplifying the localized signal up to 1000-fold by rolling-circle amplification. The signal is then visualized as discrete spots by microscope imaging.

Primary antibodies for Duolink PLA were evaluated by western blot analysis using whole-cell lysates (WCL; 50 µg per lane) and purified recombinant proteins (50 ng per lane). Rabbit anti-CFL1 (#5175 Cell Signaling Technology, Danvers, MA, USA) and rabbit anti-CFL2 (#AP20625c Abgent, San Diego, CA, USA) specifically recognized the respective recombinant proteins, as well as native proteins in WCLs at 1:1000 dilution (Figure 7a and Appendix A), while rabbit anti-CFL2 (#GTX100213 GeneTex, Irvine, CA, USA) failed to recognize recombinant CFL2. Mouse anti-CAP1 (#SAB1406999 Sigma-Aldrich, St. Louis, MO, USA) specifically recognized recombinant ΔCC-N-CAP1 and native CAP1 in WCLs at 1:500 (Figure 7a and Appendix A). However, mouse anti-CAP2 antibody (#sc-377471 Santa Cruz Biotechnology, Dallas, TX, USA) raised against amino acids 77–121 of human CAP2 failed to recognize the recombinant human ΔCC-N-CAP2 (a.a. 30–220) (Appendix A). Furthermore, while mouse anti-CAP2 (#H00010486-M01 Abnova, Taiwan) raised against GST-tagged full-length recombinant human CAP2 specifically recognized the recombinant human ΔCC-N-CAP2, staining of WCLs produced multiple major non-specific bands (Appendix A), which made this antibody unusable for PLA applications. Secondary antibodies used for western blotting analysis were anti-mouse and anti-rabbit IgG conjugated to horseradish peroxidase (#A4416 and #A0545 Sigma-Aldrich, St. Louis, MO, USA), both at 1:10,000. The signal was detected using a WesternBright Sirius chemiluminescent HRP substrate (#K-12043 Advansta, San Jose, CA, USA) in an OmegaLum G Aplegen imager (Gel Company, San Francisco, CA, USA).

The following cell lines were grown according to ATCC recommendations: HeLa, CaCo 2, LS 174T, HEK 293, HT 1080, U2OS, PANC 1, MDA-MB-231, -436, -468, SKBR 3, MCF 7, Raw 264.7, 3T3, CHOK1, CCL-39, MDCK, IEC-18, WI-38, and Hs 578T. Anti-cofilin western blots revealed that among these cell lines, Hs 578T demonstrated prominent expression of both CFL1 and CFL2 isoforms (Appendix A). Anti-CAP1 western blotting confirmed similarly high levels of CAP1 expression in Hs 578T, compared to HeLa, HT 1080, U2OS, MDA-MB-436, SKBR 3, and WI-38 (Appendix A). Therefore, the Hs 578T cell line was selected for the PLA studies.

For the Duolink in situ PLA, Hs 578T cells were plated at 30%–50% confluency on µ-slide ibiTreat 8 well plates (#80826 ibidi, Germany), allowed to attach overnight, and fixed/permeabilized for 15 min in PBS containing 4% formaldehyde and 0.1% Triton X-100. Duolink in situ PLA staining was performed according to the manufacturer’s user guide. Pairs of primary antibodies were used as followed: rabbit anti-CAP1 (#SAB1406999 Sigma-Aldrich, St. Louis, MO, USA) 1:100 with either mouse anti-CFL1 (#5175, 1:400; Cell Signaling Technology, Danvers, MA, USA) or mouse anti-CFL2 (#AP20625c, 1:100; Abgent, San Diego, CA, USA). For the negative control staining, only one primary antibody was used, followed by both PLA probes and rolling cycle amplification. Cells were counter-stained with FITC–phalloidin (Thermo Fisher Scientific; 30 nM final concentration in PBS), mounted in Duolink in situ mounting medium with DAPI (Sigma-Aldrich), and imaged using Eclipse Ti-E microscope with a 60× oil objective and DS-QiMc camera (Nikon Instruments Inc., Melville, NY, USA).

## Figures and Tables

**Figure 1 ijms-20-05647-f001:**
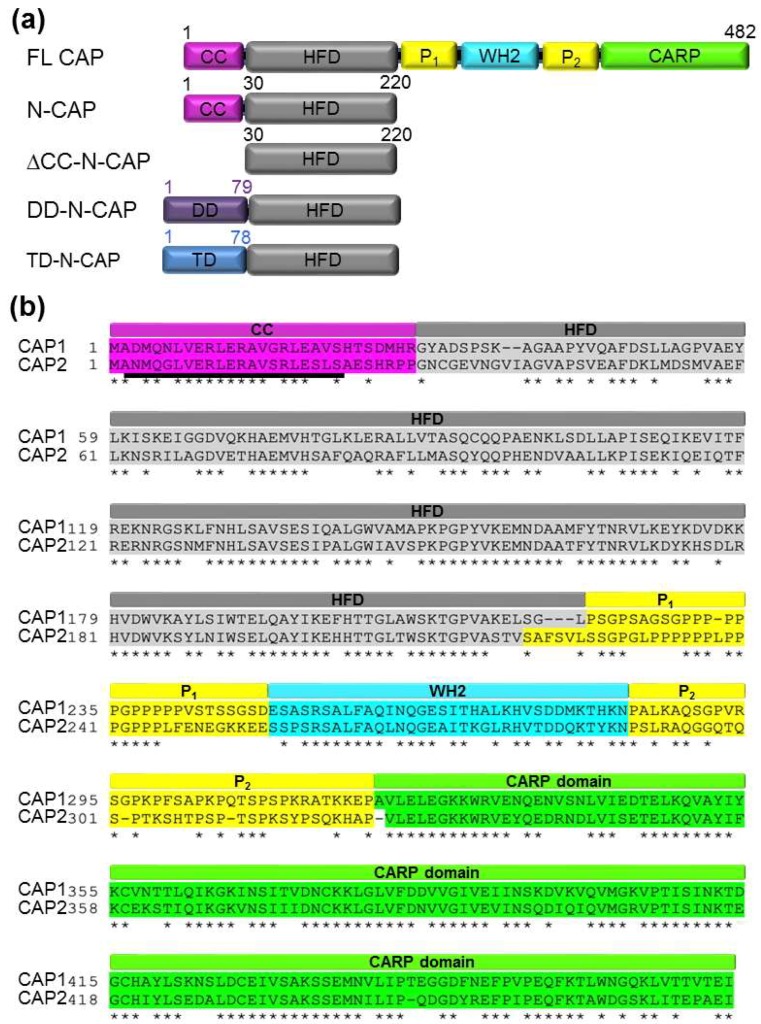
Domain structure of CAP proteins: (**a**) schematic representation of the full-length (FL) CAP domain structure and the truncation constructs used in this study. Amino acid numbering is shown for CAP1 only. CC: coiled coil domain (pink); HFD: helical folded domain (grey); P_1_ and P_2_: polyproline-rich regions (yellow); WH2: Wiskott-Aldrich syndrome protein (WASP)-homology 2 domain (cyan); CARP: C-terminal β-sheet domain (green); DD: dimerization domain; TD: trimerization domain. (**b**) Protein sequence alignment of human CAP1 and CAP2. Asterisks represent identical residues (64.1% identity) between the two isoforms. The underlined sequence with a predicted high helical propensity was used in CCBUILDER 2.0 for modeling coiled-coil oligomerization (see Figure 3). Domain abbreviation and coloring as in (a).

**Figure 2 ijms-20-05647-f002:**
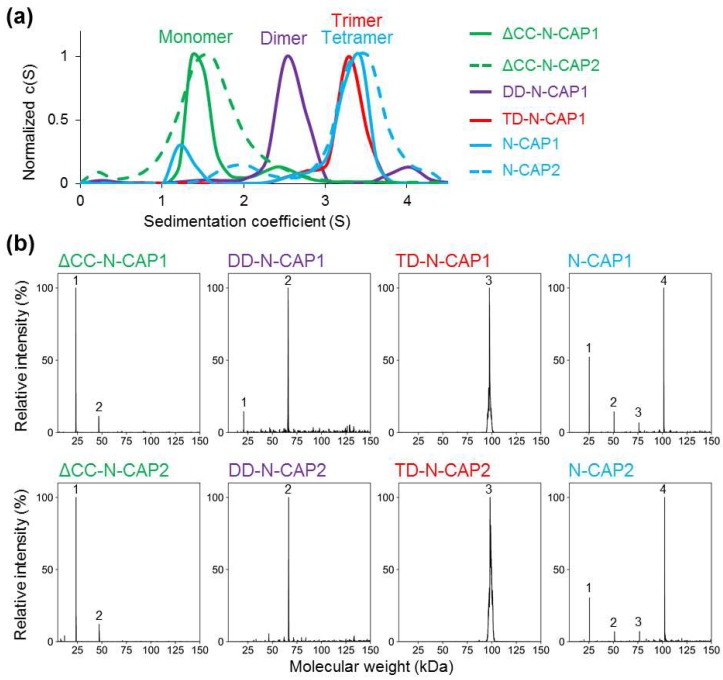
Oligomerization state of the N-terminal constructs of human CAPs: (**a**) sedimentation velocity analytical ultracentrifugation (SV-AUC) data were analyzed using SEDFIT software with a continuous sedimentation coefficient distribution model c(S). (**b**) Raw m/z data obtained by native mass spectrometry (MS) were deconvoluted with UniDec 4.0 and the molecular weights of the constructs were calculated. Numbers in the graphs indicate the oligomeric state of the CAP constructs present in solution: 1—monomer, 2—dimer, 3—trimer, 4—tetramer.

**Figure 3 ijms-20-05647-f003:**
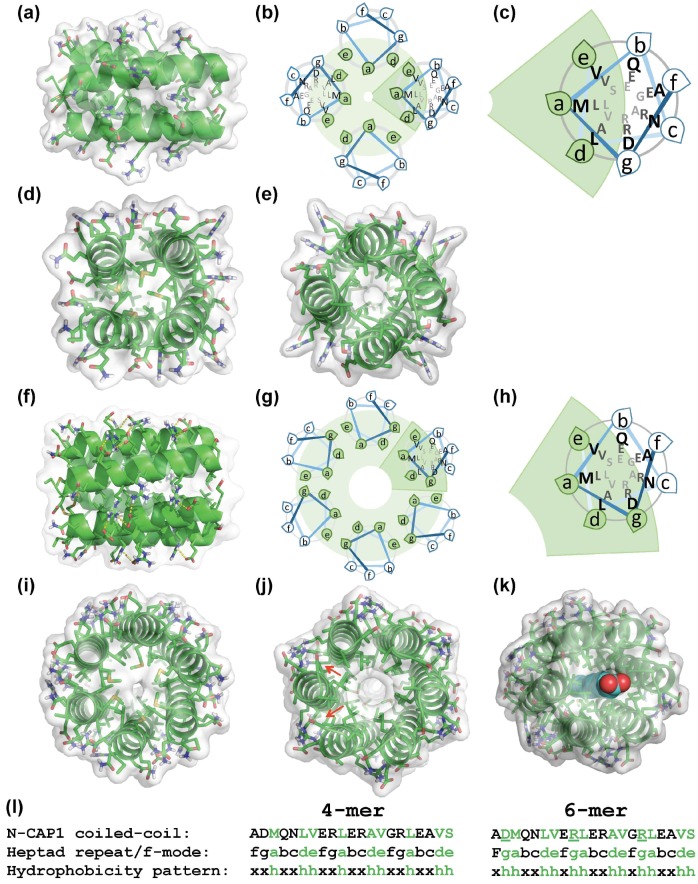
Modeling oligomerization of the N-terminal constructs of human CAPs: the CCBUILDER 2.0 web application (http://coiledcoils.chm.bris.ac.uk/ccbuilder2) was used to model the oligomeric coiled-coil structures of CAP1 (shown) and CAP2 N-terminal helices. The most stable tetramers (**a**–**e**) and hexamers (**f**–**j**) with the lowest energies are shown from three perspectives: side view (**a**,**f**), and along the helices from the N- (**d**,**i**), and C-termini (**e**,**j**). Schematic helical wheel diagrams show the relative orientations of the helices in the oligomers (**b**,**g**) and an enlarged view of the highlighted helices (c,h). Green shaded areas represent the hydrophobic cores of the oligomers. Up to four salt bridges between Arg and Asp/Glu residues contribute to the stabilization of each pair of neighboring helices (shown as yellow dotted lines in (**f**)). (**k**) Hypothetical stabilization of the hexamer structure by a fatty acid (blue) occupying the central, hydrophobic channel of Srv2/CAP (see Section 3). A low-energy CAP2 structure generated in CCBUILDER is shown. (l) The N-terminal coiled coil sequence of CAP1 (top row), with the conventional “a–f” designation of the heptad amino acids (middle row) and the hydrophobicity pattern (bottom row) characteristic of trimeric/tetrameric (xxhxxhh) and higher-order (xhhxxhh) coiled-coil structures. Residues involved in stabilization of the hydrophobic cores are designated by “h” in the hydrophobicity pattern and by green color elsewhere. Notice that in the hexamer, Cβ-Cγ atoms of the underlined Arg residues (l) contribute to the stabilization of the hydrophobic core (red arrows in (j).

**Figure 4 ijms-20-05647-f004:**
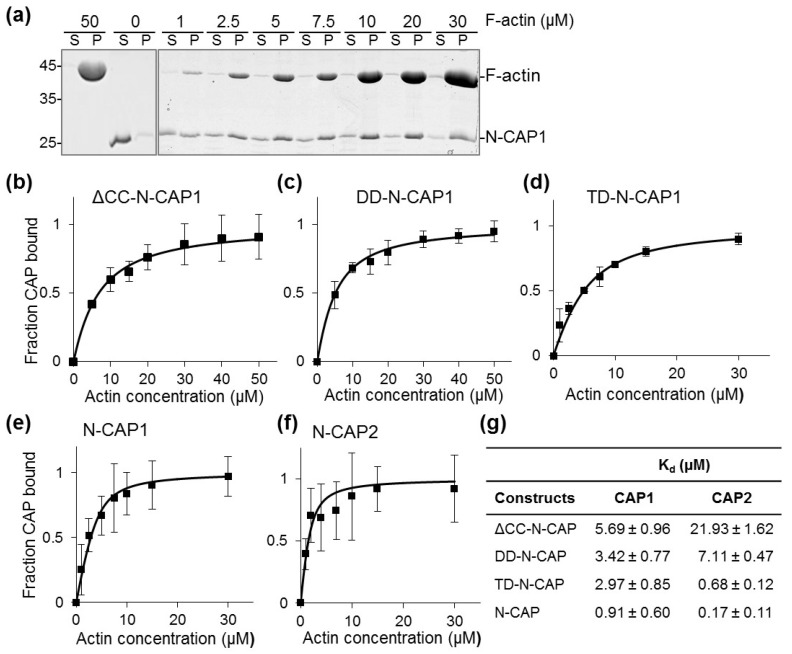
Comparison of F-actin binding affinities of human N-CAP constructs. CAP binding affinities to F-actin were analyzed by high-speed (300,000× *g*) cosedimentation: (**a**) representative sodium dodecyl sulfate polyacrylamide gel electrophoresis (SDS-PAGE) of supernatant (S) and pellet (P) fractions of N-CAP1 co-pelleted with F-actin (uncropped version of the gel is shown in Appendix A); (**b**–**f**) binding curves with error bars representing the standard errors of the mean of three independent repetitions; (**g**) K_d_ values determined by fitting the experimental data to the binding isotherm equation defined in the Section 4.4.

**Figure 5 ijms-20-05647-f005:**
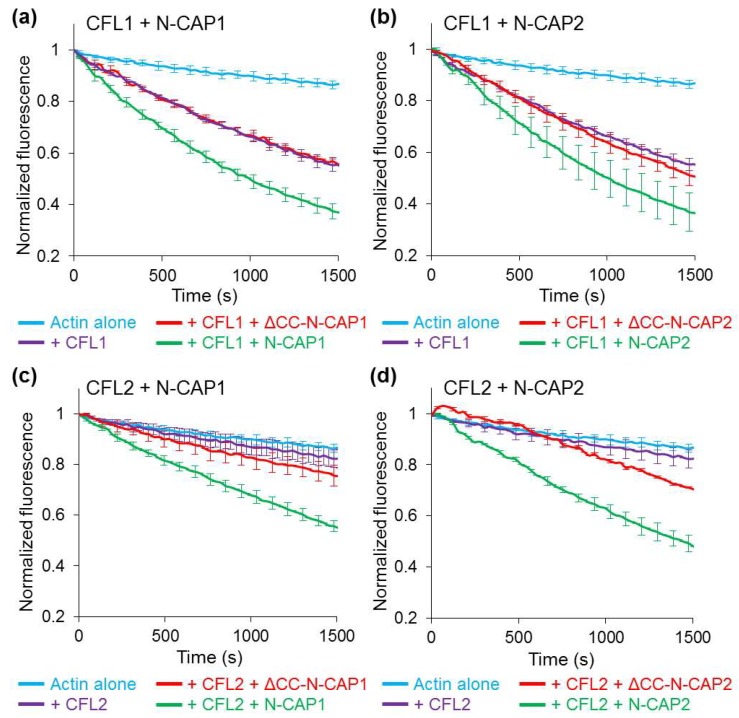
Effects of N-CAP oligomers on cofilin-mediated F-actin disassembly in bulk pyrene–actin depolymerization assays. Depolymerization of preassembled pyrene-labeled (10%) F-actin from the pointed ends (in the presence of CapZ) was initiated by the addition of a G-actin-sequestering drug latrunculin B along with 250 nM CFL1 (**a**,**b**) or CFL2 (**c**,**d**) in the absence of presence of 750 nM N-CAP1 (a,c) or N-CAP2 (b,d) oligomeric constructs. For clarity, error bars representing standard errors of the mean from three (CFL1) and four (CFL2) independent experiments are shown for every third data point.

**Figure 6 ijms-20-05647-f006:**
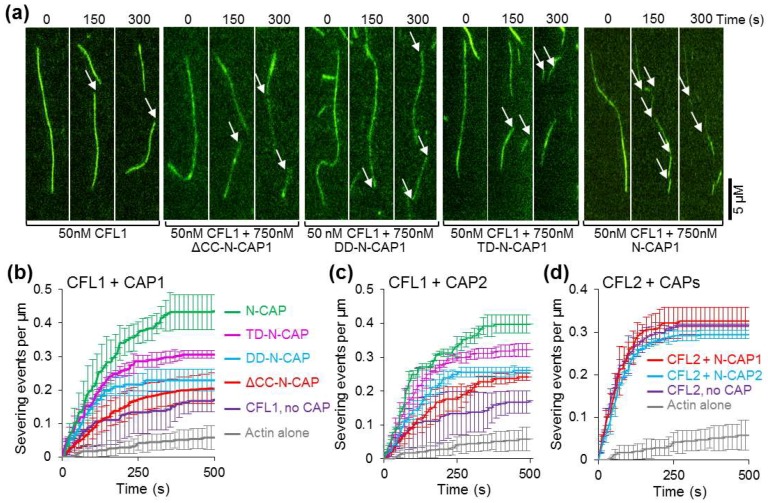
Effects of N-CAP constructs on cofilin-mediated F-actin severing observed by total internal reflection fluorescence (TIRF) microscopy. (**a**) Representative time-lapse images of Alexa 488-labeled F-actin severing upon addition of CFL1 in the absence or presence of N-CAP1 constructs. Arrows indicate severing events. (**b**–**d**) Analysis of severing activities: each data point represents the mean value of the number of severing events per micron of filament from three independent experiments (10–15 filaments per experiment). For clarity, error bars representing standard deviations of the mean are shown for every third data point.

**Figure 7 ijms-20-05647-f007:**
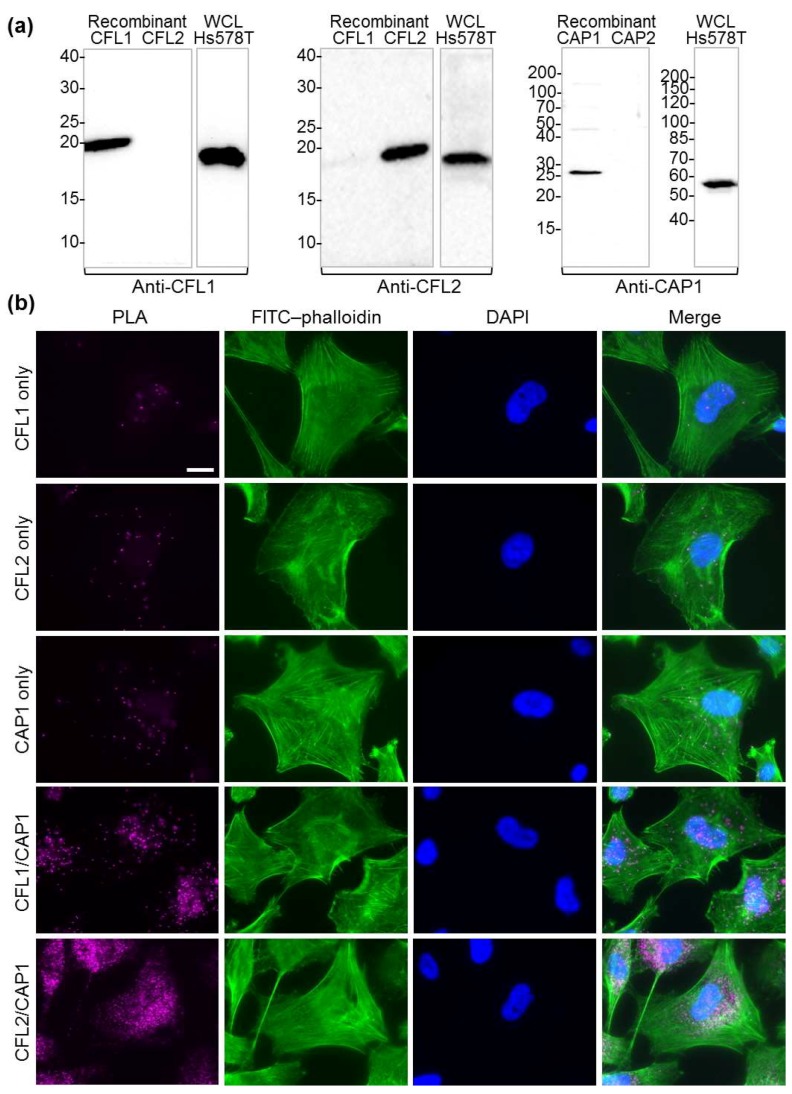
Interaction of native CAP1 with cofilin isoforms in cells: (**a**) western blot analysis demonstrates the specificity of the primary isoform-specific anti-CFL1, anti-CFL2, and anti-CAP1 antibodies used in immunofluorescence proximity ligation assay (PLA) experiments. WCL: whole-cell lysate. Uncropped versions of the blots are shown in Appendix A. (**b**) Duolink in situ PLA assay performed on Hs 578T cells, as described in the Section 4.8. Cells stained using a single primary antibody (CFL1 only, CFL2 only, and CAP1 only) are shown as negative controls. PLA signal (magenta) using pairs of CFL1/CAP1 and CFL2/CAP1 antibodies represents CAP1/cofilin interaction events. Cells were counter-stained with fluorescein isothiocyanate (FITC)–phalloidin for F-actin (green) and nuclear 4′,6-diamidino-2-phenylindole (DAPI, blue). Scale bar is 20 μm.

**Table 1 ijms-20-05647-t001:** Molecular weights (MW) of recombinant CAP constructs determined by SV-AUC and native MS. (ƒ) —frictional ratio; (S)—sedimentation coefficient.

	AUC (ƒ)	AUC (S)	AUC MW (kDa)	MS MW (kDa)	Theoretical MW (kDa)
ΔCC-N-CAP1	1.97	1.36	25.30	23.25	23.48 (monomer)
DD-N-CAP1	2.02	2.53	60.40	66.03	66.10 (dimer)
TD-N-CAP1	1.97	3.33	92.3	97.60	97.46 (trimer)
N-CAP1	2.17	3.43	92.1	101.46	101.46 (tetramer)
ΔCC-N-CAP2	2.07	1.62	23.50	23.69	23.69 (monomer)
DD-N-CAP2	1.44	3.78	66.8	66.63	66.52 (dimer)
TD-N-CAP2	1.41	4.84	92.3	98.36	98.03 (trimer)
N-CAP2	2.17	3.54	102.00	102.34	102.34 (tetramer)

**Table 2 ijms-20-05647-t002:** Predicted energies of N-terminal coiled-coil (CC) oligomers of human CAPs: provided values are means and standard deviations of 10 best scores of twenty Bristol University Docking Engine (BUDE) energies calculated by CCBUILDER 2.0.

	Oligomeric State	BUDE Energy
N-terminal CC of CAP1	Tetramer	−398.7 ± 13.7
	Pentamer	−671.4 ± 10.2
	Hexamer	−857.1 ± 73.0
	Heptamer	−916.4 ± 42.5
N-terminal CC of CAP2	Tetramer	−388.5 ± 15.4
	Pentamer	−601.2 ± 15.4
	Hexamer	−831.6 ± 65.1
	Heptamer	−878.6 ± 23.2

**Table 3 ijms-20-05647-t003:** Effects of N-CAP constructs on cofilin-mediated F-actin depolymerization rates. Initial rates of F-actin depolymerization were measured from the slopes of the pyrene–actin depolymerization curves during the first 500 seconds. Rates are expressed in nM/min as mean values with standard errors from three (CFL1) and four (CFL2) experiments.

	No CAPs	CAP1	CAP2
Actin alone	12.9 ± 2.8		
CFL1	44.0 ± 4.0		
CFL1 + ΔCC-N-CAP		44 ± 4.0	48 ± 6.9
CFL1 + DD-N-CAP		52 ± 4.0	52 ± 4.0
CFL1 + TD-N-CAP		60 ± 12	64 ± 4.0
CFL1 + N-CAP		72 ± 3.4	68 ± 14
CFL2	15.8 ± 6.5		
CFL2 + ΔCC-N-CAP		24 ± 4.0	12 ± 1.4
CFL2 + DD-N-CAP		30 ± 0.8	48 ± 3.5
CFL2 + TD-N-CAP		42 ± 6.0	44 ± 4.0
CFL2 + N-CAP		39 ± 4.0	48 ± 2.1

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
