# Peer review of "Oligomerization Affects the Ability of Human Cyclase-Associated Proteins 1 and 2 to Promote Actin Severing by Cofilins"

_ijms, 2019, doi:10.3390/ijms20225647_

Round 1
Reviewer 1 Report
The article is very interesting and written with great care. The methodology of the study were chosen correctly and proved the presented thesis. The prepared figures are legible and consistent with the description. The text is comprehensible and free of major grammatical or stylistic errors. There is only no information in the introduction about another cofilin function, which is actin transport to the nucleus. I suggest that you add 1-2 sentences about another task of this protein in the introduction.
Author Response
Reviewer 1
The article is very interesting and written with great care. The methodology of the study were chosen correctly and proved the presented thesis. The prepared figures are legible and consistent with the description. The text is comprehensible and free of major grammatical or stylistic errors. There is only no information in the introduction about another cofilin function, which is actin transport to the nucleus. I suggest that you add 1-2 sentences about another task of this protein in the introduction.
Response to Reviewer 1
We thank the reviewer for the favorable comments on our manuscript and the valuable suggestion. As suggested by the reviewer, we have included the information on the cofilin’s role in transporting actin to the nucleus and other activities of cofilin to the end of the discussion section (lines 315-325 and references 60-74).
Reviewer 2 Report
In this manuscript, the authors have assessed the oligomerizationof CAP1 and CAP2 and its impact on cofilin activity. Using analytical ultracentrifugation and native mass spectrometry, they find that both proteins (CAP1/CAP2) form tetramer. Further they analyze the F-actin binding and cofilin dependent severing activity of constructed oligomers of CAP1 and CAP2. Additionally, they show that CAP1 and CFL1/CFL2 interaction using proximity ligation assay.
The manuscript is well written and the data is thoroughly presented. The results of the manuscript are potentially interesting. The work presents significant contribution in understanding CAP1/CAP2 function. I have minor suggestions that may help improve the manuscript.
1) At several instances the letters referring part of the figure is missing. For example, Line 189: Instead of Figure 4, authors should write Figure 4 b-f.
2) Authors could perform immunoprecipitation/ in vitro pulldown experiment to show CAP1/CAP2 interaction with cofilin.
3) Including full length CAP1/CAP2 in cofilin mediated F-actin severing would strengthen the manuscript.
Author Response
Response to Reviewer 2
We thank the reviewer for the careful review of the manuscript and valuable suggestions.
1) At several instances the letters referring part of the figure is missing. For example, Line 189: Instead of Figure 4, authors should write Figure 4 b-f.
We have fixed such negligences throughout the manuscript.
2) Authors could perform immunoprecipitation/ in vitro pulldown experiment to show CAP1/CAP2 interaction with cofilin.
We agree that immunoprecipitation/in vitro pulldown experiment could be applied to demonstrate the interactions of CAPs with cofilin. This approach, however, has its limitations. Particularly, for proteins associated with the cytoskeleton, the pulldown does not allow to differentiate between direct and indirect (cytoskeleton-mediated) interactions. Therefore, we decided to take advantage of a more advanced and direct assay (i.e., rolling cycle amplification-based proximity ligation) to demonstrate the interaction and localization of cofilins and CAPs.
3) Including full length CAP1/CAP2 in cofilin mediated F-actin severing would strengthen the manuscript.
We agree that including the full-length CAPs could strengthen the manuscript. We attempted to purify the full-length protein from human embryonic kidney cells (HEK-293 cells), but the yield was insufficient to conduct the desired experiments. Given that only the N-terminal half of CAP1/2 is involved in potentiating the severing activity of ADF/cofilins, we consider that the goal of testing the role of oligomerization on this activity has been achieved by using the recombinant N-CAP constructs.